# Machine Learning for Predicting Biologic Agent Efficacy in Ulcerative Colitis: An Analysis for Generalizability and Combination with Computational Models

**DOI:** 10.3390/diagnostics14131324

**Published:** 2024-06-21

**Authors:** Philippe Pinton

**Affiliations:** Clinical and Translational Sciences, International PharmaScience Center Ferring Pharmaceuticals, Amager Strandvej 405, 2770 Kastrup, Denmark; philippe.pinton@ferring.com; Tel.: +45-28-78-74-50

**Keywords:** biologic agents, machine learning, artificial intelligence, computational models, prediction, ulcerative colitis, precision medicine

## Abstract

Machine learning (ML) has been applied to predict the efficacy of biologic agents in ulcerative colitis (UC). ML can offer precision, personalization, efficiency, and automation. Moreover, it can improve decision support in predicting clinical outcomes. However, it faces challenges related to data quality and quantity, overfitting, generalization, and interpretability. This paper comments on two recent ML models that predict the efficacy of vedolizumab and ustekinumab in UC. Models that consider multiple pathways, multiple ethnicities, and combinations of real-world and clinical trial data are required for optimal shared decision-making and precision medicine. This paper also highlights the potential of combining ML with computational models to enhance clinical outcomes and personalized healthcare. Key Insights: (1) ML offers precision, personalization, efficiency, and decision support for predicting the efficacy of biologic agents in UC. (2) Challenging aspects in ML prediction include data quality, overfitting, and interpretability. (3) Multiple pathways, multiple ethnicities, and combinations of real-world and clinical trial data should be considered in predictive models for optimal decision-making. (4) Combining ML with computational models may improve clinical outcomes and personalized healthcare.

Key Questions

Question

What are the benefits of using machine learning in predicting clinical outcomes?

Answer

Machine learning can provide precision, personalization, efficiency, and automation, and it can improve decision support in predicting clinical outcomes. It can analyze large and diverse datasets, identify intricate patterns, automate data analysis, and assist clinicians in making evidence-based predictions.

Question

What are the challenges in using machine learning for predicting clinical outcomes?

Answer

Challenges in machine learning predictions include low data quality and data scarcity, overfitting, and poor generalizability and interpretability. Inadequate or biased data may lead to inaccurate predictions. Overfitting occurs when models learn noise from training data, and some models lack interpretability, which is crucial for clinicians to trust and understand the predictions.

Question

What factors should predictive models notably consider for optimal decision-making in ulcerative colitis?

Answer

Predictive models of ulcerative colitis should consider multiple pathways, multiple ethnicities, and combinations of real-world and clinical trial data. By incorporating these factors, the models can ensure optimal shared decision-making and precision medicine in the treatment of ulcerative colitis.

Question

Why is it important to validate machine learning models across different populations?

Answer

Validating machine learning models across different populations, especially for treating ulcerative colitis, is important to ensure the translatability and generalizability of the models. Differences in efficacy and the response to biologic agents may occur between ethnicities, and validating models across diverse populations can help to determine their applicability and effectiveness in different patient groups.

## 1. Introduction

Ulcerative colitis (UC), a chronic inflammatory bowel disease, poses significant therapeutic hurdles. While biologic agents have revolutionized UC treatment, optimizing their use remains crucial to surpass efficacy limitations. Machine learning (ML) and computational modeling present promising avenues for addressing these challenges. By analyzing clinical trials and real-world data, incorporating multiple omics and markers, improving diagnostic accuracy, predicting patient responses, and personalizing therapy, ML can enhance treatment outcomes and reduce reliance on steroids. Additionally, computational modeling enables us to simulate disease progression and predict both short-term and long-term treatment responses, leading to transformative approaches in UC management.

## 2. Machine Learning

ML is an interdisciplinary field that combines principles from computer science, mathematics, and statistics. It involves the development of algorithms and models that allow computers to learn from data and make predictions or decisions without requiring explicit programming to that end [1].

Conventional risk stratification methods have traditionally relied on statistical techniques. However, in the last decade, advanced computational approaches—such as machine learning—have gained prominence for risk prediction and prognosis across various conditions. Specifically, artificial intelligence methods for analyzing medical images (e.g., endoscopy, radiology, histology) have garnered significant attention in managing inflammatory bowel disease (IBD). Nonetheless, whether machine-learning-based prediction models using routine clinical data or clinical trial data provide any clear advantages over conventional risk prediction methods remains debated, especially when it comes to personalized medicine in daily practice.

The main benefits of ML in predicting clinical outcomes include (1) precision and personalization, (2) efficiency and automation, and (3) improved decision support. The main pitfalls and challenges of ML in this task include (4) low data quality and data scarcity, (5) overfitting and low generalizability, and (6) lack of interpretability [2,3,4,5,6,7,8,9]. These characteristics are described as follows:
(1)ML models can analyze large and diverse datasets and identify intricate patterns that may not be apparent to human clinicians. Personalized predictions allow for tailoring treatments to individual patients and improve clinical outcomes.(2)ML automates the data analysis and reduces manual effort and time. Predictive models can quickly process vast amounts of information, thereby aiding clinical decision-making.(3)ML assists clinicians in providing evidence-based predictions. It can complement medical expertise and enhance diagnostic accuracy and treatment planning.(4)ML models require high-quality data. Inadequate or biased data can result in inaccurate predictions. Data scarcity may limit model performance, especially for rare conditions.(5)Overfitting occurs when a model learns noise from the training data, leading to poor generalization to new data. Balancing model complexity and generalizability is crucial.(6)Some ML models (e.g., deep neural networks) lack interpretability. Clinicians must trust and understand ML predictions; improving the interpretability of ML models is thus necessary.


ML has recently emerged as a powerful tool for predicting outcomes and personalizing treatments for inflammatory bowel disease (IBD), including ulcerative colitis (UC). Its use in predicting the efficacy and safety of therapeutic agents is expected to help practitioners with their prescriptions. In fact, ML techniques allow practitioners to aggregate and analyze vast quantities of data as well as identify and quantify features that may lead to novel insights into disease management. Various studies have incorporated clinical, laboratory, and omics data to predict significant outcomes in IBD [10]. These data include hospitalization records, corticosteroid use, biologic responses, and refractory disease after colectomy. In addition, encouraging results from various medical fields (e.g., histopathology, endoscopy, and radiology) support the use of ML image analysis to advance outcome prediction for IBD [11].

ML models using baseline patient data alone have achieved an area under the receiver operating characteristic curve of 0.62 in predicting the remission for vedolizumab in UC. When the data from week 6 were included, the area under the curve improved to 0.73. Remarkably, patients predicted to be remitters had a higher probability of achieving corticosteroid-free endoscopic remission [11]. The Paddington international virtual chromoendoscopy score histologic remission index and a related artificial intelligence system have been found to be a simple scoring system and predictor of histological remission in UC, respectively [12]. Moreover, ML models based on serological markers have demonstrated high accuracy in predicting UC relapse. These models may help determine personalized treatment plans [13].

For this opinion paper, we conducted a search in PubMed for eligible studies published between 1 January 2014 and 31 December 2023. Our focus was on articles describing prediction modeling using machine learning techniques, specifically related to UC and biologic agents. To be eligible, publications needed to describe the development or validation of at least one multivariable prediction model aimed at individualized efficacy outcome prediction. We did not impose restrictions based on study design, data source, or patient-related health outcomes. We considered a study to involve machine learning when a non-regression statistical technique was used to create or validate a prediction model. Excluded were publications reporting single predictors, tests, or biomarkers, as well as those using machine learning solely for image or signal enhancement or relying exclusively on genetic traits or molecular markers as predictors. We also excluded systematic reviews, methodological articles, conference abstracts, and publications unavailable through our institution. Our search was limited to studies involving human participants and articles written in English.

Additionally, as we wanted to reflect on the opportunities of developing models with one or multiple pathways, and of using real-world data or clinical trials data, we limited ourselves to biologic agents that were included in modeling exercises with such features. Vedolizumab and ustekinumab were the only biologic agents matching the mentioned criteria [14,15,16].

This opinion paper focuses then on two recent models and suggests different or complementary approaches for predicting the efficacy and safety of biologic agents for treating UC [14,15]. It includes a generative artificial intelligence and computational modeling combination, serving to exploit the potential of artificial intelligence but still benefit from the traceability back to causality that the computational modeling approach provides.

## 3. Example of Efficacy Prediction Using ML: Vedolizumab and Ustekinumab for Treating UC

Morikubo et al. [14] and Miyoshi et al. [15] recently proposed ML models for predicting the efficacy of ustekinumab and vedolizumab in patients with UC. They developed and tested prediction models using real-world clinical data as the baseline from Japanese patients who received ustekinumab or vedolizumab for treating UC. Their models achieved high predictive values of 88.9% and 92.3%, respectively. However, for the same two biologic agents in Crohn’s disease (CD), the model for vedolizumab could not be applied to ustekinumab and required the selection of specific features using the random forest algorithm [16]. The difference in the mechanism of action between ustekinumab and vedolizumab could be the main reason why the model initially developed for vedolizumab does not work for ustekinumab. This is well-supported by the inclusion in the models of the fraction of monocytes (vedolizumab) and of lymphocytes (ustekinumab).

The model developed by Morikubo et al. [14] could have also benefited from a longer follow-up period to evaluate the long-term efficacy of biologic agents and their impact on disease progression and remission. This is a critical aspect to envision, knowing that a loss of response may occur after several months of treatment. CM can help, as shown by Venkatapurapu et al. [16].

The vedolizumab clinical decision support tool (VDZ-CDST) was developed to predict the treatment efficacy in patients with refractory CD who were intolerant to anti-TNF (tumor necrosis factor) drugs. The predictive capabilities of this tool were assessed using vedolizumab and ustekinumab. Although successful prediction of clinical and steroid-free clinical remissions was obtained for vedolizumab, the tool did not demonstrate the same efficacy for ustekinumab in this patient population. The specific ML approach used to create VDZ-CDST was not mentioned by Alric et al. [17]. In addition, the reason for ustekinumab not demonstrating the same efficacy as vedolizumab was not stated. However, we can make the following speculations based on the available information:
(1)Mechanisms of action: Vedolizumab and ustekinumab have different mechanisms of action, which may contribute to the varying responses in patients with refractory CD. This is broadly in line with previous works [18].(2)Patient heterogeneity: CD is a heterogeneous disease, and the varied patient profiles in the study population may have led to substantial differences in the responses to biologic agents.(3)Sample size and study design: The study sample size (insufficient statistical power) and design (study duration) possibly undermined the results.


The latter are well-identified issues with ML studies. Numerous investigations into ML-based prediction models reveal suboptimal methodological rigor and a substantial risk of bias. Factors contributing to this bias include participant exclusions, small sample sizes, inadequate handling of missing data, and a failure to address overfitting. To enhance the utility of supervised ML models in clinical practice and minimize research inefficiencies, efforts should focus on improving the study design, execution, reporting, and validation [19].

## 4. Analysis and Discussion

We recommend taking up the developments by Morikubo et al. [14] and Miyoshi et al. [15] given their innovative and useful predictive tools, which may help clinicians select the optimal treatment for patients with UC in the near future [20]. We provide perspectives regarding the generalizability and applicability of their models to other populations or settings. The predictive tools developed by Morikubo et al. [14] and Miyoshi et al. [15] may support decision-making and precision medicine. These tools may assist clinicians and patients by providing personalized insights, optimizing treatment choices, and improving patient outcomes based on individual characteristics and preferences. Tools for the optimal prediction of IBD should balance complexity and interpretability.

Integrating diverse omics strategies within a network framework can enhance our understanding of the pathogenesis involved in IBD. By analyzing various molecules—such as genomic, transcriptomic, proteomic, microbiome, epigenetic, and metabolomic data—simultaneously, we can develop multiomics models that provide valuable insights. These models may help identify disease subgroups, optimize therapy regimens, and discover promising predictive biomarkers, ultimately facilitating early diagnosis [21].

Although incorporating multiple immune pathways can enhance accuracy, the models must remain clinically applicable and transparent for shared decision-making. To enhance choice and decision, the models underlying predictive tools can consider a ubiquitous path instead of a specific target, such as vedolizumab and the α4β7 receptor. For example, STAT 3 modulation intersects with several therapeutic agents that have distinct signatures correlated with clinical benefits [22,23].

Rogers et al. [24] reported the potential of a pharmacological model with dynamic quantitative systems for considering the immune system in IBD. Although modeling IBD with multiple pathways and layers is highly complicated and requires large datasets, a simulation model of various immune networks allows for the consideration of different therapeutic strategies based on the IBD pathoetiology and on the patient’s phenotype at the treatment onset.

The model proposed by Rogers et al. [24] can simulate both CD and UC using shared underlying biological mechanisms but distinct initial conditions, which more accurately represent the specific subtype of IBD. The model has been designed in a modular manner, incorporating each biological interaction as a Michaelis–Menten-type reaction. This modular structure allows for easy addition, removal, or modification of interactions based on new biological insights.

Numerous approved or under-development drugs target the mechanisms, e.g., recruitment of leukocytes to sites of inflammation or Th cell differentiation, considered in the model of Rogers et al. [24]. Ustekinumab is one of those, and the combination of this approach with the one proposed by Morikubo et al. is an opportunity [25].

Tools like the abovementioned ones represent a step forward in precision medicine. Currently, computational models (CMs), platforms, digital twins, and quantum computers can enable the implementation of such solutions [16,26,27,28,29,30].

### 4.1. ML and CMs

ML and CMs present complementary characteristics, and their combination may lead to powerful hybrid approaches that leverage their strengths. Combining these two approaches requires a thoughtful design, a full understanding of the problem domain, and careful integration. These considerations are essential to balance data-driven learning and domain-specific knowledge. Precision medicine can benefit from such a combination for improving clinical outcomes and personalized healthcare.

ML is a specialized branch within the broader field of artificial intelligence. It is focused on creating algorithms that can learn from data without explicit programming. Popular ML models include decision trees, naïve Bayes, random forests, and support vector machines. These models analyze features in the available data to create decision boundaries for prediction or classification. ML models learn from data by adjusting their internal parameters based on observed patterns. Based on training data, they generalize to make predictions on previously unseen data. ML is well-suited for processing unstructured data.

CMs encompass a wide range of approaches, including traditional mathematical models, simulations, and rule-based systems. These models are designed to represent and simulate real-world phenomena and are often based on mathematical equations or logical rules. In fact, they are typically based on mathematical formulations or predefined rules. CMs do not learn from data but rely on explicit instructions. Hence, CMs are suitable for handling structured data.

Weather forecasting provides a fair illustration of the potential benefit of such a combination. Similar behaviors can be applied to UC and biologic agents.

#### 4.1.1. Traditional Mathematical Model (CM)

In this case, for the weather forecast, we create a mathematical model based on physical principles. We consider factors like temperature, humidity, wind speed, and pressure. The model incorporates differential equations that describe how these variables interact. By solving these equations, we simulate how weather conditions evolve over time. This approach relies on predefined rules derived from scientific knowledge. For instance, the Navier–Stokes equations describe fluid flow, which can be applied to air movement in the atmosphere. By solving these equations, we predict wind patterns and temperature changes. CMs like this are suitable for structured data because they rely on explicit mathematical formulations.

For UC, similar to weather models, CM models rely on predefined rules based on scientific knowledge. In UC treatment, e.g., we can consider immune pathways (such as cytokine signaling, e.g., TNF-α, IL-23), disease duration (chronicity affecting treatment response), and genomic factors (including HLA variants that influence susceptibility and response to biologics). Differential equations describe how these variables interact, reflecting disease progression and treatment outcomes. By solving these equations, we simulate treatment responses over time. Clinicians can use CMs to predict the efficacy of specific biologic agents based on individual patient characteristics.

#### 4.1.2. Machine Learning Approach

ML models, on the other hand, learn from historical data. We collect past weather data (temperature, humidity, etc.) and corresponding outcomes (rain, sunshine, etc.). Using algorithms like decision trees, neural networks, or regression, the ML model identifies patterns and relationships in the data. Once trained, the ML model can predict tomorrow’s weather based on the current conditions (input features). However, ML models may struggle with extreme events or sudden changes if not enough relevant data are available.

For UC, we collect, e.g., data on past biologic responses (remission rates, mucosal healing) and patient-specific features (genetic markers, disease duration). Algorithms like decision trees or neural networks identify patterns. The ML model predicts the efficacy of a biologic agent based on current patient data. However, ML models may struggle with extreme cases (e.g., refractory UC) due to limited relevant data.

To improve accuracy, we can combine both approaches, and keeping the weather forecast illustration: Use the CM to simulate large-scale weather patterns (e.g., global air circulation). Then, fine-tune the predictions using ML based on local data (e.g., historical weather records for a specific city). This hybrid approach leverages the strengths of both CMs (domain-specific knowledge) and ML (data-driven learning).

In summary, CMs encompass various techniques, including traditional mathematical models and simulations. While CMs rely on explicit rules, ML learns from data. Combining these approaches can enhance precision and personalized outcomes in fields like weather forecasting or healthcare.

ML and CMs can be combined in multiple ways [31,32].
(1)Ensemble methods:Boosting: Boosting algorithms, such as AdaBoost or XGBoost, combine multiple weak ML models, often decision trees, to create a strong ensemble. These models mitigate the errors of their constituent models to improve the overall performance.Stacking: Stacking involves training multiple ML models (base learners) and using their predictions as inputs for another model (meta-learner). The meta-learner learns to appropriately combine the base model outputs.(2)Physics-informed ML:By combining ML techniques with domain-specific knowledge, such as fluid dynamics principles, encoded in a CM, ML models can predict turbulence based on data, whereas computational fluid dynamics models provide physical insights. Integrating both approaches can enhance prediction accuracy.(3)Data preprocessing: ML techniques preprocess data before feeding them into CMs.ML can manage missing data and perform feature engineering and noise reduction, thereby improving the quality of CM inputs.(4)Hybrid models: Models that blend ML and computational components can be constructed. For example, an artificial neural network (ML) combined with a differential equation solver (CM) can be used to model chemical reactions.(5)Uncertainty quantification:ML models often lack uncertainty estimates.ML predictions can be combined with CMs for including uncertainty bounds (e.g., Bayesian models).(6)Transfer learning:An ML model can be trained on a task to then be fine-tuned for a specific computational problem.Transfer learning can proceed from an ML model to CM.

### 4.2. Randomized Clinical Trials and Real-Word Datasets

Combining individual patient data collected from randomized controlled clinical trials with real-world datasets can enhance the development of predictive tools and guide the integration of multiple treatments in shared decision-making and precision medicine for IBD. Clinical trial datasets have become more accessible via open-access platforms, and real-world-evidence trial datasets have gained accuracy, as demonstrated by Morikubo et al. [14] and Miyoshi et al. [15].

Platforms such as Vivli can help design models for multiple treatments using predictive tools. Vivli provides access to anonymized individual-level data collected during clinical trials. An example is a predictive model for vedolizumab using its pivotal phase 3 clinical program in UC. The VARSITY and tofacitinib trial datasets were integrated, allowing the model to identify patients’ responses to adalimumab, tofacitinib, and vedolizumab [33].

### 4.3. Ethnicity Considerations

Using a dataset collected from a population of a single country to train and validate a model causes uncertainty in the translatability to other countries and populations. However, existing clinical trial data on vedolizumab in UC seem to confirm translatability to the general public. The pharmacokinetic parameters in Asian and non-Asian patients with UC are similar, and the results of a clinical trial performed in Japanese patients with UC confirmed the suitability of a specific dose of vedolizumab for application in patients worldwide [34,35]. In contrast, translatability is not obvious for ustekinumab, and the performance of the model developed by Morikubo et al. [14] for non-Asian patients with UC is uncertain owing to some differences in the efficacy of ustekinumab between Asian and non-Asian patients. According to the UNIFI program, a global phase 3 trial of ustekinumab for treating UC, the clinical remission rates at weeks 8 and 44 were lower in East Asian patients than in the overall population; however, the differences were not statistically significant [36,37,38]. In detail, a post hoc subpopulation analysis of the UNIFI program showed that clinical remission rates during both the induction and maintenance phases were lower in patients from East Asia (i.e., China, Japan, Korea, and Taiwan) than in the overall population. The clinical remission rates at week 8 were 11.4% (6 mg/kg) and 11.1% (130 mg) in East Asian patients and 15.6% (6 mg/kg) and 15.5% (130 mg) in the overall population, and those at week 44 were 47.6% (90 mg q12 w) and 19.2% (90 mg q8w) in East Asian patients and 38.4% (90 mg q12 w) and 43.8% (90 mg q8w) in the overall population [38]. As the differences in the efficacy of ustekinumab between Asian and non-Asian patients with UC were not statistically significant, they may have occurred by chance, whereas the safety profiles of ustekinumab are likely similar.

While current evidence suggests that ustekinumab is generally effective and safe for treating UC in patients from different regions and ethnicities, the observed variations in the response rates between Asian and non-Asian patients may bias prediction, if not anticipated, when developing a model. The differences may be attributable to various factors, such as genetics, including the genetic expression of IL-12 and IL-23 receptors, pharmacokinetics, disease characteristics, or concomitant medications, which may influence the response to ustekinumab. Population pharmacokinetics and exposure–response analyses conducted in patients with CD showed a higher clearance of ustekinumab in Asian patients than in Caucasian patients. This effect was observed after controlling for weight [39].

In recent years, various studies have demonstrated that numerous IL-23 receptor polymorphisms are correlated with the susceptibility of UC and CD. For example, the dominant model of the rs10889677 polymorphism is associated with the risk of developing UC in Asians but not in Caucasians [40]. Likewise, differences potentially exist when considering single-nucleotide polymorphisms of the IL-12B gene [41,42].

Currently, no clinical predictive tools dedicated to ustekinumab that consider ethnic differences are available. Most existing studies are based on clinical and biochemical variables, such as disease location, prior anti-TNF exposure, serum albumin, and fecal calprotectin. The available tools have been developed and evaluated in different populations and settings. However, no tool has specifically addressed the differences between Asian and non-Asian patients with UC. Therefore, assessing existing models in diverse regions or across ethnicities and comparing the results with those of Japanese cohorts is a valuable future research direction.

### 4.4. Dosage Alternatives

The model’s applicability to patients who receive different doses or regimens of ustekinumab should be evaluated considering the variety of dosing and monitoring strategies for patients with UC. In the UNIFI program, the patients received an intravenous induction dose of ustekinumab based on their body weight (either 130 mg or 6 mg/kg), followed by a subcutaneous maintenance dose of 90 mg every 8/12 weeks. In clinical practice, some patients require dose escalation, interval shortening, or therapeutic drug monitoring to achieve or maintain the clinical response. Therefore, a model for patients receiving different doses or regimens of ustekinumab should be evaluated, and the impact of drug exposure or pharmacokinetic parameters on the model performance should be explored.

## 5. Conclusions

In conclusion, a predictive tool should involve different pathways to help physicians achieve the most suited and beneficial decision for the patient and should consider different omics and their impact on accuracy and translatability. A combination of ML and CMs can be helpful. The parameters identified by Morikubo et al. [14] have the potential to align with the model developed by Rogers et al. [24].

The contributions of Morikubo et al. [14] and Miyoshi et al. [15] to personalized medicine for UC are highly valuable and deserve continued development. We expect that their models will be further and prospectively validated and refined in different populations and settings toward optimizing the use of ustekinumab and other biologic agents for treating patients with UC.

Predictive models, combining machine learning and computational modeling, offer personalized treatment optimization. Clinicians can tailor therapies based on patient-specific data, detect disease progression early, optimize dosages, assess safety profiles, and predict treatment responses. These models enhance shared decision-making, inform clinical trial design, and integrate real-world evidence. However, challenges include model validation and clinician training. Overall, this approach holds promise for precision medicine in UC management.

Precision medicine requires balancing innovation (treatment, diagnostic, generative artificial intelligence, quantum computers, cloud), equity, individual benefit and societal impact, ethics, and scalability. It needs evidence from clinical trials and real-world practice. It can also incorporate lifestyle, environmental, and social determinants of health. Ensuring affordability and cost-effectiveness will determine its sustainability. As we move forward, collaboration among researchers, clinicians, policymakers, payers, and patients will shape the future of precision medicine.

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
