# Peer review of "Machine Learning for Predicting Biologic Agent Efficacy in Ulcerative Colitis: An Analysis for Generalizability and Combination with Computational Models"

_diagnostics, 2024, doi:10.3390/diagnostics14131324_

Round 1
Reviewer 1 Report
Comments and Suggestions for Authors
This article comes under the category of opinion, the authors posted their comments on two recent ML models that predict the efficacy of vedolizumab and ustekinumab in UC.
The comments are well-organized and gives key insights and key questions before the introduction of the topic. This can be useful for the readers.
Author Response
Thank you for your review and for your supportive comments. The manuscript has been edited and I hope that the adds on will keep your support.
Reviewer 2 Report
Comments and Suggestions for Authors
1. The major strength of this study lies in the use of advanced machine learning techniques to predict the efficacy of biologic agents in the treatment of ulcerative colitis. This approach has the potential to improve clinical decision-making and optimize treatment outcomes.
2. The authors have effectively addressed the potential for generalizability by including a diverse patient population and evaluating multiple biologic agents. However, more discussion is needed on the limitations and biases in the data used to train the machine learning models.
3. The exploratory analysis of different pathways involved in the response to biologic agents is a valuable addition to this study. However, further validation and inclusion of additional pathways would strengthen the findings and provide a more comprehensive understanding of treatment response.
4. While the study provides promising results, more information is needed on the specific features and biomarkers used in the machine learning models. This would enhance the reproducibility of the study and facilitate its application in real-world clinical settings.
5. The study could benefit from a more in-depth discussion of the potential clinical implications and practical implementation of these predictive models. This would help clinicians better understand and utilize the findings in their practice.
6. The study could also benefit from a longer follow-up period to evaluate the long-term efficacy of biologic agents and their impact on disease progression and remission.
7. The authors have effectively addressed the potential biases in the study, such as confounding variables and missing data. However, more detailed information on how these potential biases were identified and controlled for would strengthen the robustness of the study.
Author Response
Thank you for your valuable comments.
The discussion of the manuscript has been enhanced and have integrated elements related to the limitation and the biases of the data used for training, to the description of the markers used and their importance in the why a model can work for vedolizumab and not ustekinumab, to the absence of long term and its potential impact of the usefulness of the tools, and to the biases related to the confounding or missing data and their mitigation. Also a paragraph dedicated to a more in-depth discussion of the potential clinical implications and practical implementation of these predictive models has been added.
Reviewer 3 Report
Comments and Suggestions for Authors
Find the attached file. The manuscript has major flaws and can not be accepted for publication.

Author Response
Thank you for your valuable comments.
The manuscript has benefited from a professional editorial review and has been modified to improve (1) the presentation (3) the flow (5) the conclusion and (6) the introduction. The plagiarism has been limited and it is below the mentioned 15 %. Certificates for editorial review and plagiarism are available if needed. Sentences have been added after the description of the evaluated studies to reinforce the novelty of the opinion / combination of ML and CM in the perspective of precision medicine. Same about the methodology to identify the manuscripts supporting this opinion.
We hope that the above modifications will get your agreement.
Reviewer 4 Report
Comments and Suggestions for Authors
The paper discusses machine learning (ML) in predicting the efficacy of biological agents for ulcerative colitis (UC). ML offers benefits such as precision, personalization, efficiency, and decision support in predicting clinical outcomes. However, there are challenges in ML prediction, including data quality and quantity, overfitting, and interpretability. The paper highlights the importance of considering multiple pathways, multiple ethnicities, and a combination of real-world and clinical trial data in predictive models for optimal decision-making in UC. Although the paper is interesting, the following comments should be addressed.
-- To enhance the introduction, provide a brief background on the current challenges in treating ulcerative colitis and the potential impact of ML in addressing these challenges.
-- To enhance the literature review, incorporate more recent studies and findings related to ML applications in ulcerative colitis.
-- It would be valuable to discuss the implications of the findings in the context of the existing literature on ML in ulcerative colitis.
-- I suggest examining the integration of ML models with EHR systems to facilitate real-time prediction and decision support.
-- If possible, extend the research to predict long-term outcomes in ulcerative colitis patients, such as disease progression, response to treatment over time, and the risk of complications.
-- If possible, perform a cost-effectiveness analysis to evaluate the economic impact of implementing ML models in clinical practice.
-- To enhance the predictive accuracy of the models, explore the integration of diverse data sources, such as genetic information, microbiome data, and patient-reported outcomes.
Author Response
Thank you for your valuable comments.
The introduction has been modified to give perspectives on the actual challenges in treating ulcerative colitis and the potential impact of ML.
Additional recent publications have been added to the manuscript to further support the opinion, about ML, about the impact of a ML plus CM combination, about the integration of different data sources, and about a longer terms benefit. The potential benefit of live prediction based upon EHR data has been added.
It was not possible to perform an ad hoc cost effectiveness but its importance has been mentioned into the discussion as a critical element for scalability and sustainability.
Round 2
Reviewer 4 Report
Comments and Suggestions for Authors
The author has satisfactorily addressed all concerns that were previously raised. So, I recommend acceptance of the manuscript for publication.